# Characteristics of Pathogenic *Escherichia coli* Associated with Diarrhea in Children under Five Years in Northwestern Ethiopia

**DOI:** 10.3390/tropicalmed9030065

**Published:** 2024-03-21

**Authors:** Berihun Mossie Mulu, Mequanint Addisu Belete, Tiliksew Bialfew Demlie, Habtamu Tassew, Tesfaye Sisay Tessema

**Affiliations:** 1Department of Veterinary Science, College of Agriculture and Environmental Sciences, Bahir Dar University, Bahir Dar 79, Ethiopia; berihunmossie@gmail.com (B.M.M.); tiliksew.bialfew@bdu.edu.et (T.B.D.); habtamu.tassew@bdu.edu.et (H.T.); 2Department of Veterinary Laboratory Technology, College of Agriculture and Natural Resource, Debre Markos University, Debre Markos 269, Ethiopia; 3Institute of Biotechnology, Addis Ababa University, Addis Ababa 1176, Ethiopia; tesfaye.sisayt@aau.edu.et

**Keywords:** children, diarrheagenic *Escherichia coli*, resistance, virulent genes, Awi zone

## Abstract

Diarrheagenic *Escherichia coli* (DEC) are the leading cause of infectious diarrhea and pose a significant global, regional, and national burden of disease. This study aimed to investigate the prevalence of six DEC pathotypes in children with diarrhea and determine their antibiotic resistance patterns. Samples from 107 diarrheagenic children were collected and processed for *Escherichia coli* (*E. coli*). Single-plex PCR was used to detect target virulence genes as well as characterize and categorize DEC pathotypes. Antibiotic resistance patterns were determined by the Kirby–Bauer disk diffusion method. *E. coli* was detected in 79 diarrheal stool samples, accounting for 73.8% of the samples collected. Additionally, 49.4% (39 out of 79) of the isolates harbored various typical virulence factors. Results revealed six pathotypes of virulence: enterotoxigenic *E. coli* (ETEC) (53.8%), enteropathogenic *E. coli* (EPEC) (12.8%), enteroaggregative *E. coli* (EAEC) (10.3%), Heteropathotypes (7.8%), Shiga toxin-producing *E. coli* (STEC), and enterohemorrhagic *E. coli* (EHEC) (7.7% each). The isolates exhibited high antibiotic resistance against trimethoprim/sulfamethoxazole (82.1%), amoxicillin (79.5%), ampicillin (74.4%), gentamicin (69.2%), and streptomycin (64.1%). An overall occurrence of 84.6% of multiple-drug resistance was observed in the isolates, with resistance ranging from three to four antibiotic classes. Our findings revealed a high level of pathogenic *E. coli* that were highly resistant to multiple categories of antibiotics among children in the Awi zone. These findings highlight the potential role of pathogenic *E. coli* in childhood diarrhea in tropical low-resource settings and underscore the need for continued research on the characteristics of pathogenic and antibiotic-resistant strains.

## 1. Introduction

*Escherichia coli* was previously recognized as a harmless commensal and normal intestinal microorganism that provided some benefit to its hosts. However, the ability of *E. coli* to acquire virulent genes allows them to be highly diverse and adaptable pathogens [1]. The genes encoding *E. coli* virulence factors are located in plasmids, in large genome regions called pathogenicity islands (10 to 200 kb), or in integrated bacteriophages [2]. It can cause a wide range of diseases from the gastrointestinal tract to extraintestinal sites such as the urinary tract, bloodstream, and central nervous system [3].

The acquisition of various virulence genes has resulted in the formation of specific pathotypes involved in diarrheal diseases called diarrheagenic *E. coli* (DEC) [4]. DEC isolates are classified based on the phenotypic traits and the presence of individual and combined virulence factors: enterotoxigenic *E. coli* (ETEC), enteroinvasive *E. coli* (EIEC), enteroaggregative *E. coli* (EAEC), enteropathogenic *E. coli* (EPEC), enterohemorrhagic *E. coli* (EHEC), Shiga toxin-producing *E. coli* (STEC), diffusely adherent *E. coli* (DAEC) [5,6].

EIEC is characterized by the ability to invade and destroy enterocytes. The invasion of EIEC is mediated by a gene on a large invasion plasmid (*plnv*) and bacterial chromosome [7]. ETEC produces heat-labile (LT) and heat-stable (ST) toxins encoded by the *eltAB* and *estAB* genes [8,9]. In the pathogenesis of EAHC, aggregative adherence fimbriae (*AAFs*) play a role in adherence and epithelial damage. The *pAA* plasmid contains the master regulator *aggR*, which regulates the expression of genes and classifies EAEC into typical and atypical strains [10]. EPEC pathotypes are characterized by the ability to cause attaching and effacing (A/E) lesions and are identified by the existence of particular virulence factors including intimin (encoded by the *eae* gene) and bundle-forming pili (*Bfp*). It is subdivided into typical and atypical strains based on the presence of the *E. coli* adherence factor (EAF) plasmid. EPEC strains that possess the virulence plasmid EAF are considered typical [11]. The major virulence factors in STEC infection are potent Shiga toxins encoded by the *stx1* and *stx2* genes. The subgroup EHEC is determined by the acquisition of an additional intimin (*eae*) gene and an entero-hemolysin (*hlyA*) gene [12]. Watery diarrhea and dysentery with blood and mucus are the typical characteristics of DEC infections in young children [13].

There are more than 1.7 billion cases of childhood diarrhea disease globally, resulting in around half a million (525,000) deaths among children under five each year [14]. Indeed, diarrheal diseases are among the leading causes of child morbidity and mortality in low–middle-income countries (LMICs), accounting for over 90% of deaths in children under five years of age. The 2009–2018 demographic and health survey (DHS) of 34 Sub-Saharan African countries revealed a 15.3% overall prevalence of diarrhea among children under five [15,16]. DEC pathotypes, particularly EAEC, EPEC, and ETEC, are among the primary agents of moderate–severe diarrhea, responsible for about 30–40% of cases of diarrhea episodes and 70,000 diarrhea deaths of children younger than five years [17]. In Ethiopia, deaths due to diarrhea have shown a decline over time following the implementation of various strategies like Community-Led Total Sanitation and Hygiene (CLTSH) and Sustainable Development Goals (SDGs). However, it remains the greatest threat to the health of children under five years with a pool prevalence of 22% [18,19,20]. For example, 25,139 deaths due to diarrhea were recorded in 2019 among children under five [21].

Gram-negative bacteria can acquire and transmit antimicrobial resistance (AMR), thereby becoming resistant to multiple classes of antibiotics, making it difficult to effectively treat common infections. Multidrug-resistant Enterobacteriaceae, including *E. coli*, has been recognized by WHO as a critical priority pathogen that poses a serious threat to human health [22]. In low-resource settings, the emergence and spread of AMR have been associated with a link between humans, food-producing animals, and the environment. However, there is a scarcity of microbiological, epidemiological, and social science research at the community and population level to fill a large gap in the knowledge of AMR [23,24].

Several previous studies have shown the prevalence and distribution of DEC as a cause of diarrhea in children under five years old in different areas of Ethiopia [25,26,27,28,29]. However, research on the molecular-based detection and pathotyping of *E. coli* isolates in the Awi zone, Northwest Ethiopia, is limited. Awi is a zone in the country that has one of the highest rates of diarrhea in children under the age of five [30,31,32]. Thus, there is a need to investigate whether the DEC is responsible for the high prevalence of childhood diarrhea in this area. Therefore, the overall aim of this study was to characterize pathogenic *E. coli* associated with intestinal infections in children in Northwest Ethiopia.

## 2. Materials and Methods

### 2.1. Description of the Study Area

This study was conducted in selected hospitals (Injibara General Hospital, Dangila, and Agew Gimjabet Primary Hospitals) in the administrative zone of Awi of the Amhara Regional State in Northwest Ethiopia. The global positions of Injibara, Dangila, and Agew Gimjabet on the Google Earth tool (2020) are indicated by their latitude and longitude of 10°57′ N, 36°56′ E; 11°16′ N, 36°50′ E; and 10°51′ N, 36°52′60′′ E, with an approximate elevation of 2560, 2137, and 2329 m above sea level, respectively (Figure 1).

### 2.2. Study Design and Subjects, and Sampling

A cross-sectional study was conducted from November 2021 to May 2022. Convenient and purposive sampling methods were used to select study sites and collect samples. Children under five years of age who visited those three hospitals with acute diarrhea were included in this study. Acute diarrhea is defined as the passage of loose or watery stools usually occurring at least three times in 24 h or three or more days and lasts no longer than 14 days [33]. However, all children who received antibiotic therapy within the previous two weeks and older than five years were excluded from this study. The questionnaires were designed to evaluate information about anamnesis of children. Approximately 25 g of diarrheal stool samples was were collected by laboratory technicians from the respective hospitals using sterile stool cups containing buffered peptone water.

### 2.3. Bacterial Isolation and Identification

Each diarrheal sample was directly cultured on Eosin Methylene Blue Agar (EMB) (Oxoid, Basingstoke, Hampshire, UK). Colonies showing green metallic sheen characteristics on EMB agar were considered presumptive *E. coli* isolates. Next, pure colonies stored in nutrient broth were further characterized for biochemical activity using biochemical tests: indole, methyl red, Vogues, Proskauer, and citrate utilization (IMViC). Bacterial isolates exhibiting an IMViC (+ + − −) pattern were confirmed as *E. coli* isolates [34].

### 2.4. DNA Extraction and Polymerase Chain Reaction

DNA from *E. coli* was extracted using the boiling method from overnight Tryptone Soya Broth (TSB) cultures. The Nanodrop 2000 spectrophotometer (Thermo Scientific, Wilmington, DE, USA) was used to determine the concentration and purity of DNA. The extracted DNA was stored at −20 °C and used for the molecular detection of the virulence genes. Each PCR reaction was performed in a total of 25 µL mixtures containing 0.5 µL of each primer (0.2 µM, Bioneer, Daejeon, South Korea), 2.5 µL of the PCR buffer (10X, Solis Bio Dyne, Tartu, Estonia), 1 µL of dNTP (100 mM, HiMedia, Mumbai, India), 2 µL of MgCl_2_ (25 mM, Solis Bio Dyne), 0.5 µL of the Taq polymerase enzyme (5 U/µL, Solis Bio Dyne), 15 µL of nuclease-free water, and 3 µL of template DNA.

PCR amplifications were performed on the A300 fast gradient thermal cycler (LongGene Scientific Instruments, Hangzhou, China). The reaction mixtures were amplified with an initial denaturation at 95 °C for 3 min., followed by 35 cycles of a denaturation step at 95 °C/40 s, annealing (Table 1)/40 s, and extension at 72 °C for 1 min, and the final extension step was at 72 °C/10 min. The PCR products were resolved using 1.5% agarose gel electrophoresis (*w*/*v*), stained with ethidium bromide in 1× TAE buffer (40 mM Tris, 20 mM acetic acid, 1 mM EDTA, pH 8.3). The size of the PCR amplicon was verified by comparison with a 100 bp DNA ladder (HiMedia, Mumbai, India). The gel result was visualized and photographed under a Gel Doc EZ Gel Documentation System (Bio-Rad, Hercules, CA, USA). The primer gene sequences, target genes, respective pathotypes of *E. coli*, and amplicon size in the base pairs are shown in Table 1.

### 2.5. Susceptibility Testing

Antimicrobial susceptibility testing (AST) was carried out using ten different antibiotics from six classes comprising fluoroquinolones [ciprofloxacin (5 µg), nalidixic acid (30 µg)]; penicillins [amoxicillin (10 µg), ampicillin (10 µg)]; aminoglycosides [gentamycin (10 µg), streptomycin (10 µg)]; folate antagonists [trimethoprim/sulfamethoxazole (25 µg), trimethoprim (5 µg)]; cephalosporins [cefotaxime (30 µg)]; and tetracyclines [tetracycline (30 µg)] (Oxoid, Hampshire, United Kingdom) on a 100 mm plate. The disk diffusion method was used for the AST of all isolates following Clinical Laboratory Standards Institute (CLSI) guidelines [44]. Mueller Hinton agar plates (HiMedia, Mumbai, India) were inoculated with a suspension of a pure culture equilibrated to 0.5 McFarland standards. The results were recorded after 24 h of incubation at 37 °C, and the diameter of the inhibition zone was interpreted as susceptible, intermediate, and resistant following the breakpoints. Isolates with nonsusceptibility to at least one agent in three or more antimicrobial categories were considered to have multidrug resistance (MDR) [45].

### 2.6. Data Analysis

All data were entered and analyzed using Microsoft Office Excel^®^ 2019 (Microsoft Corporation, Redmond, WA, USA) and Graph Pad Prism (v9.5.1) (GraphPad Software, Boston, MA, USA). Descriptive statistics were computed by determining absolute frequencies along with their respective percentages. The occurrence of diarrheagenic *E*. *coli*, the detection of the virulence gene, and the extent of antibiotic resistance were compared among study subjects in selected hospitals.

## 3. Results

### 3.1. Escherichia coli Isolation

In total, 107 stool samples were collected from children with diarrhea in three selected public or government hospitals. The samples were collected from three geographically distinct areas, as follows: Injibara town (*n* = 43), Dangila town (*n* = 38), and Agew Gimjabet town (*n* = 26) (Table 2).

The conventional analysis revealed that *E. coli* was recovered in 79 (73.8%) diarrheal stool samples. *E. coli* isolates were more prevalent in males (51.4%) than in females (48.1%). Bacterial infection was high in the age group 25–60 months (63.3%), followed by the medium age group 7–24 months (27.8%) (Table 2). The rates for *E. coli* infection were 35.4%, 34.2%, and 7.6% in the groups of children fed mothers’ milk and supplemented with complementary foods, fed solid food, and exclusively breastfed, respectively. Basic information on diarrheal children and the distribution of *E. coli* and DEC concerning various variables is displayed in Table 2.

### 3.2. Occurrence of Virulence Genes and DEC Pathotypes

Of 79 isolates, 39 (49.4%) carried one or more putative genes associated with virulence and were confirmed as DEC. The highest detection rate was recorded in diarrhea stool samples from Dangila Primary Hospital (41%), followed by Injibara General Hospital (31%), and Agew Gimjabet Primary Hospital (28%). The detection of DEC is more frequent in male (58.9%) children who used tap water sources (61.5%) and those with a history of contact with animals (56.4%) (Table 2). The presence of the *st* gene was observed in 33.3% of DEC isolates, followed by the *lt* gene, which was detected in 12.8% of cases. Notably, 7.7% of isolates harbored both *lt* and *st* genes. Additionally, the *eae* and *aatA* genes were identified in 12.8% and 10.3% of the DEC isolates, respectively. The *stx2* gene was detected in 7.7% of isolates. Heteropathotypes, characterized by the presence of a combination of various virulence genes from two pathotypes (*st*/*hlyA*, *eae*/*st*/, and *aatA*/*st*), were detected at an overall rate of 7.8%. The *st* gene was the most prevalent while the genes *bfpA*, *ial*, and *daaE* were not found in any of the isolates (Figure 2 and Appendix A).

Overall, six different DEC pathotypes were identified based on the combination of virulence genotypes: ETEC, aEPEC, EAEC, Heteropathotypes, STEC, and EHEC. Pathotype detection rates of 51.3%, 25.6%, and 23.1% were recorded at the Dangila Primary, Injibara General, and Agew Gimjabet Primary Hospitals, respectively. All pathotypes were detected in the hospitals sampled except for STEC and Heteropathotypes, which were absent at the Agew Gimjabet Primary Hospitals (Figure 2). The most common pathotype was ETEC, presented in 53.8% of cases, followed by EPEC in 12.8%. To a lesser extent, the STEC and EHEC strains were detected with an equal percentage of 7.7%. Heteropathotypes (ETEC-EHEC, aEPEC-ETEC, EAEC-ETEC) were detected in 7.8% of DEC isolates, with an equal percentage (2.6%) of gene combinations (Figure 2).

### 3.3. Antimicrobial Resistance in DEC Isolates

Figure 3 shows the distribution of the phenotypic antibiotic resistance profiles of DEC isolates. High levels of antibiotic resistance were found against trimethoprim/sulfamethoxazole (82.1%), amoxicillin (79.5%), ampicillin (74.4%), gentamicin (69.2%), and streptomycin (64.1%). On the other hand, DEC isolates from diarrheal children revealed a high level of sensitivity to nalidixic acid (87.2%), cefotaxime (76.9%), ciprofloxacin (66.6%), tetracycline (61.5%), and trimethoprim (56.4%) (Figure 3). Furthermore, 84.6% of the isolates exhibited MDR and showed ten antimicrobial resistance patterns. The predominant MDR pattern in DEC isolates was resistance to three antibiotic categories (84.9%), followed by resistance to four classes of antibiotics (15.1%). The less frequently observed phenotype was resistance to seven antibiotic agents (3%) (Table 3).

## 4. Discussion

The identification and characterization of DEC pathotypes are critical to understanding the underlying mechanisms of *E. coli*-related diarrheal diseases. However, the incidence and impact of DEC infections in low-resource settings, including Ethiopia, remain poorly understood primarily due to the absence of coordinated epidemiological surveillance systems and advanced laboratory services [46]. The present study identified DEC pathotypes in diarrheal children. We determined each virulence factor and antibiotic resistance profile using a combination of molecular techniques and a phenotypic analysis. Our findings revealed that 39 (49.4%) DEC possess various virulence genes, including *st*, *lt, aatA*, *eae*, *stx2*, and *hlyA* that enable them to cause diarrheagenic illness. The proportion of DEC detected in this study was higher than the 20.6% to 36.4% reported from Iran [47], Kenya [48], and Egypt [49]. A similar finding with a higher frequency of pathogenic *E. coli* was documented in Nigeria [46], Gabon [50], Iraq [51], Guinea Bissau [52], and Sudan [53].

In this study, an elevated occurrence of DEC infection was observed in children who were fed solid food in comparison to those exclusively fed mothers’ milk. The difference in the rate of infection could be attributed to the protective capacity of maternal immunity transmitted through breastfeeding [54,55]. In particular, there is a prevailing notion that breastfeeding plays a crucial role in the protection of children against *E. coli* infection due to the presence of antibodies specifically targeting virulence factors [56]. Observations in previous studies demonstrate that components in the milk provide protective factors that prevent STEC attachment to the intestine and block the path of pathogenesis [57]. The impact of breastfeeding on the morbidity and mortality associated with diarrhea in infants and children has been well documented in recent studies, particularly in developing countries. These studies have consistently demonstrated that partial or no breastfeeding is correlated with an increased incidence of diarrhea in young children [58]. On the contrary, the introduction of solid and complementary foods has been implicated as a source of contamination potentially to the development of intestinal infections in infants and children under five years old. Supportive findings from low-resource settings indicate that children fed with complementary foods often exhibit high levels of contamination with pathogenic microbes [59,60,61].

The intestinal tract of animals serves as a prominent natural reservoir for human DEC [62]. The frequent occurrence of DEC among children who have a history of contact with animals in the present study suggests direct or indirect contact with an animal reservoir or with their feces as the possible route of transmission for intestinal pathogenic *E. coli*. A comprehensive meta-analysis of demographic health survey data from 30 Sub-Saharan African countries examined the association between child health outcomes and household ownership of livestock, revealing animals as a significant risk factor for both acute enteric infections and overall mortality in children [63]. There is formative evidence from Ethiopia that underscores the pivotal role of animal feces and animal husbandry practices as an important source of the bacterial contamination and transmission pathway to infants [64]. The study of Belete et al. [65] revealed a frequent detection of DEC isolates among children who have a history of contact with diarrheic calves, highlighting the significance of these interactions.

The DEC pathotypes, specifically ETEC, aEPEC, EAEC, STEC, EHEC, and Heteropathotypes, were responsible for the diarrhea of children in the study area. ETEC was the most common pathotype detected (53.8%) among DEC. These strains are the most common cause of acute childhood diarrhea in LMICs, where hygiene standards are often deplorable [66,67]. The current result showed a higher detection rate than reports from India, with percentages of 5.7% and 6% [68,69]. In our study, 33.3% of ETEC isolates produced heat-stable toxins (STs), 12.8% produced heat-labile toxins (LTs), and 7.7% produced both enterotoxins. These findings are consistent with those of a study conducted in Kenya [70] and Indonesia [71], where ST toxins were the most common, followed by LT, and both enterotoxins were the least common. In contrast, studies from Iran [72] and Bolivia [73] found that LT toxins were detected at a higher rate than ST toxins.

aEPEC was the second most common pathotype, accounting for 12.8% of all cases. All EPEC strains detected in this study were *eae*-positive and thus classified as atypical EPEC. Our findings are in line with those of Borujerdi et al. [47], who reported that the EPEC strain was the second most common pathotype and only possessed eae genes. The current finding, however, contradicts reports from developed and developing countries demonstrating an increase in the detection rate of aEPEC strains over tEPEC strains [49,74]. The failure to identify the *bfp* gene is most often because *bfp* is the structural gene that encodes the bundle-forming pilus (bfp), and these fimbriae are produced only under specific cultural conditions [75].

EAEC is a significant emerging pathogen associated with persistent diarrhea in children in developing countries. It causes watery diarrhea without blood and intermittent abdominal cramps with no fever [76]. In this study, the proportion of EAEC was found to be 10.3%, which contradicts findings in Nigeria [77], China [78], and Qatar [79]. However, these findings are close to the proportions (3.8%) and (6.5%) reported by Mabika et al. [50] in Gabon and Shatub et al. [51] in Iraq. The disparity in EAEC prevalence reported in these studies may be due to differences in sample types and geographical areas.

The present study detected a low (7.7%) proportion of STEC that harbored *stx2* genes. This finding agrees with a recent study in China [79], which reported a percentage of 3.7%. The results of the current study were slightly higher than the reports of Huang et al. [80], with a detection rate of 0.4%. Furthermore, Bonkoungou et al. [81] found 1% of STEC strains in human diarrheic stool samples in Ouagadougou, Burkina Faso. We detected 7.7% of EHEC strains, slightly higher than a study by Shatub et al. [51] that reported a detection rate of 4.3%. In contrast, Al-Dulaimi et al. [82] and Abbasi et al. [17] revealed a comparable prevalence of EHEC, with 7.8% and 9.3%, respectively.

*Escherichia coli* exhibits remarkable genomic plasticity, facilitating the emergence of new strains capable of harboring virulence genes with the characteristics of two or more DEC pathotypes. This enables *E. coli* to adapt to different environments and host immune responses, increasing its pathogenic potential [83]. Heteropathotypes with a combination of two virulence genes accounting for 7.8% were identified in this study. The result was higher than a study conducted in India [84], which found a detection rate of 3.8%. In other studies, hybrid pathotypes were identified among children suffering from diarrhea with a corresponding rate of 6% in Pakistan [76], 3.9% in Nigeria [85], and 14% in Ethiopia [65].

The current study has revealed distinctive distribution patterns of antimicrobial susceptibility among hospitals, suggesting the presence of varying local therapeutic practices by healthcare providers, influenced by several factors. Prescription practices play a crucial role in determining the types and frequencies of antibiotic agents used, thereby impacting the development of antibiotic resistance. The preference for antibiotic usage patterns is primarily influenced by educational qualifications, experience, sources of updated knowledge, and the practice setting of the clinician. Another significant factor that influences prescription patterns is the presence of hospital protocols and local guidelines for antibiotic use. In low-resource settings, access to appropriate diagnostic tests for pathogen identification and drug availability are identified as the major factors influencing prescription decisions [86,87].

*Escherichia coli* has been shown to play a significant role in the emergence of antimicrobial resistance due to its capacity to accumulate resistance genes. On the other hand, *E. coli* can also serve as an indicator organism to estimate the burden and trend of antimicrobial resistance in humans and animals [88,89]. Antibiotics including cotrimoxazole, amoxicillin, and cephalosporins are among the commonly used antibiotics for the treatment of bacterial diarrhea in Ethiopia [90,91]. In the present study, DEC of children with diarrhea showed higher resistance to trimethoprim/sulfamethoxazole (82.1%), amoxicillin (79.5%), ampicillin (74.4%), and gentamicin (69.2%). This finding agrees with the resistance reported recently in Kenya [48]. In Burkina Faso, Bonko et al. [92] also reported that isolates from children under five exhibited a high resistance rate to trimethoprim-sulphamethoxazole (100%), ampicillin (100%), ciprofloxacin (71.4%). Another study in the Maasai community of Kenya revealed high sensitivity to ampicillin, chloramphenicol, gentamycin, tetracycline, and trimethoprim/sulfamethoxazole [93].

Multidrug-resistant bacteria pose a significant threat to global public health. This highlighted that high resistance not only reduces treatment options but also profoundly impairs the efficacy of managing bacterial diseases. In addition, these MDR isolates can transfer antimicrobial resistance characteristics to susceptible (non-resistance) strains through genetic change. In this study, 84.6% of *E. coli* isolates were MDR with three or more antibiotic classes, which agree with the findings of Estrada-Garcia et al. [94]. A probable reason for such widespread resistance could be the extensive use of antibiotics in the treatment of diarrheal disease in children, often without performing antibiotic sensitivity tests or frequent prescription practices from professionals, or the acquisition of certain common antibiotic resistance genes through horizontal gene transfer from donor bacteria, phages, or free DNA [95]. This calls for further understanding of the molecular mechanisms of the development of antibiotic resistance in *E. coli* and other related bacteria.

In this study, we examined the phenotypic, genotypic, and antibiotic resistance profiles of DEC isolated from children under five suffering from diarrhea. The study further reported a high prevalence of multiple antibiotic-resistant diarrheagenic *Escherichia coli* (MRDEC) strains. These strains pose a significant threat to public health in terms of disease management difficulties and incidence of infection. The findings could provide essential information for the prevention and treatment of diarrhea in low-income settings. The limitation of this study is that results from a cross-sectional study and purposive sampling may not accurately reflect the overall prevalence and identify the primary-causing co-pathogens in children under five in the area. Further studies using case–control studies and probability sampling methods are needed to provide a more comprehensive understanding of the prevalence and causative agents of diarrheal diseases. Additionally, our research is only restricted to samples from children with diarrhea, and suggests various source samples, such as food products, environmental samples, and animal samples, to clarify the sources of contamination in the future. Further research using molecular typing methods should also explore the genetic similarities among DEC isolates and transmission dynamics in household and hospital settings.

## 5. Conclusions

The result of this study indicates that pathogenic *E. coli* play a role in child diarrhea in Northwest Ethiopia. ETEC and EPEC pathotypes were detected more frequently and can be a major source of diarrhea in low-income settings. This study revealed high rates of resistance to the class of folate antagonists, penicillins, and aminoglycosides, alongside alarming rates of MRDEC strains. Routine surveillance systems, the development of new antibiotics, and the exploration of alternative treatment strategies that emphasize antisense oligonucleotide and bacteriophage therapy technologies could offer a relevant solution to combating antibiotic-resistant *Escherichia coli* strains. In conclusion, continued epidemiological and genetic studies are required to better understand emerging strains of pathogenic *E. coli* and the risk factors and genetic mechanisms associated with antibiotic resistance.

## Figures and Tables

**Figure 1 tropicalmed-09-00065-f001:**
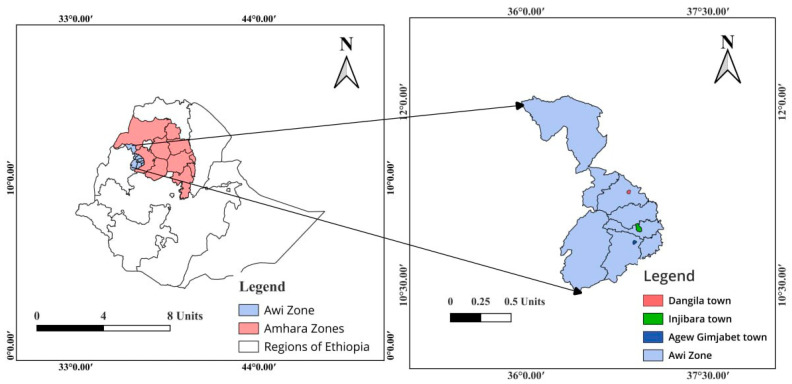
Map of the study area. The map was created by using Esri’s ArcGIS^®^ 10.8 desktop GIS software.

**Figure 2 tropicalmed-09-00065-f002:**
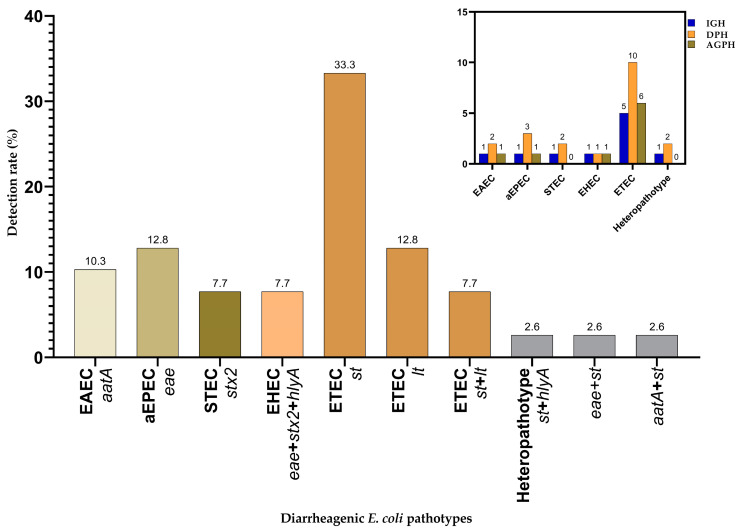
Frequency of virulent genes and diarrheagenic *E. coli* pathotypes detected in children with diarrhea and hospitals. IGH: Injibara General Hospital, DPH: Dangila Primary Hospital, AGPH: Agew Gimjabet Primary Hospital, *eae*: attaching–effacing gene, *stx2*: Shiga-like toxin II gene, *hlyA*: *E. coli* hemolysin gene, *aatA*: aggregative adherence gene, *lt*: heat-liable toxin gene, *st*: heat-stable toxin gene.

**Figure 3 tropicalmed-09-00065-f003:**
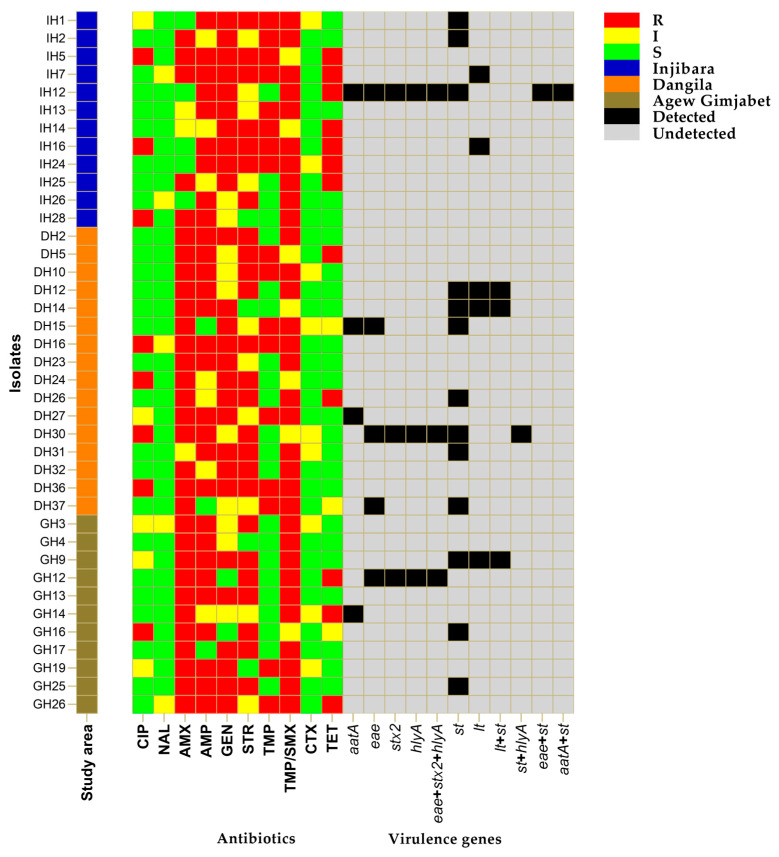
Heat map showing the distribution of antimicrobial susceptibility and virulence profiles in 39 pathogenic *E. coli* isolates exhibited from samples of diarrheic children. CIP: ciprofloxacin, NAL: nalidixic acid, AMX: amoxicillin, AMP: ampicillin, GEN: gentamicin, STR: streptomycin, TMP: trimethoprim, TMP/SMX: trimethoprim/sulfamethoxazole, CTX: cefotaxime, TET: tetracycline, *eae*: attaching–effacing gene, *stx2*: Shiga-like toxin II gene, *hlyA*: *E. coli* hemolysin gene, *aatA*: aggregative adherence gene, *lt*: heat-liable toxin gene, *st*: heat-stable toxin gene.

**Table 1 tropicalmed-09-00065-t001:** Oligonucleotide primers used in single-plex PCR for amplification of diarrheagenic virulence genes.

Primer	Nucleotide Sequence (5′ to 3′)	Target Gene	Pathotypes	Annealing Temperature	Amplicon Size (bp)	Reference
EVT1 EVT2	F: CAACACTGGATGATCTCAGC R: CCCCCTCAACTGCTAATA	*stx2*	STEC/EHEC	55 °C	350	[35]
EAE-1 EAE-2	F: AAACAGGTGAAACTGTTGCC R: CTCTGCAGATTAACCTCTGC	*eae*	EPEC/EHEC	55 °C	490	[36]
ST1 ST2	F: TTT ATT TCT GTA TTG TCT T R: GCAGGATTACAACACAATTC	*st*	ETEC	55 °C	294	[37]
IAL F IAL R	F: CTGGATGGTATGGTGAGG R: GGAGGCCAACAACATTATTTCC	*ial*	EIEC	55 °C	320	[38]
LT1 LT2	F: GGCGACAGATTATACCGTGC R: CCGAATTCTGTTATATATGTC	*lt*	ETEC	48 °C	696	[39]
EAEC F EAEC R	F: CTGGCGAAAGACTGTATCAT R: CAATGTATAGAAATCCGCTGTT	*aatA*	EAEC	48 °C	630	[40]
daaE1 daaE2	F: GAACGTTGGTTAATGTGGGGT R: TATTCACCGGTCGGTTATCAG	*daaE*	DAEC	47 °C	542	[41]
BFPF BFPR	F: AATGGTGCTTGCGCTTGCTGC R: GCCGCTTTATCCAACCTGGTA	*bfpA*	EPEC	57 °C	324	[42]
EHEC F EHEC R	F: ACGATGTGGTTTATTCTGGA R: CTTCACGTCACCATACATAT	*hlyA*	EHEC	45 °C	167	[43]

**Table 2 tropicalmed-09-00065-t002:** Background information of 107 diarrheal children and the occurrence of DEC with different factors in three public hospitals in the Awi zone, Ethiopia.

Variables	No. of Tested (*N* = 107)			Culture-Positive (*N* = 79)			PCR-Positive (DEC) (*N* = 39)			Total		
	IGH *n* = 43	DPH *n* = 38	AGPH *n* = 26	IGH *n* = 32	DPH *n* = 29	AGPH *n* = 18	IGH *n* = 12	DPH *n* = 16	AGPH *n* = 11	Total^t^ *n* = 107	Total^c^ *n* = 79	Total^p^ *n* = 39
Sex												
Female	20 (38.5)	18 (34.6)	14 (26.9)	15 (39.5)	17 (44.7)	6 (15.8)	5 (31.3)	8 (50)	3 (18.8)	52 (48.6)	38 (48.1)	16 (41)
Male	23 (41.8)	20 (36.4)	12 (21.8)	17 (41.5)	12 (29.3)	12 (29.3)	7 (30.4)	8 (34.8)	8 (34.8)	55 (51.4)	41 (51.9)	23 (58.9)
Age class												
0–6 months	6 (37.5)	4 (25)	6 (37.5)	3 (42.8)	2 (28.6)	2 (28.6)	2 (66.6)	0	1 (33.3)	16 (14.9)	7 (8.9)	3 (7.7)
7–24 months	20 (58.8)	12 (72.8)	2 (5.8)	15 (68.2)	7 (31.8)	0	3 (33.3)	6 (66.6)	0	34 (31.8)	22 (27.8)	9 (23.1)
25–60 months	17 (29.8)	22 (38.6)	18 (31.6)	14 (28)	20 (40)	16 (32)	7 (25.9)	10 (37)	10 (37)	57 (53.3)	50 (63.3)	27 (69.2)
Feeding												
Mother’s milk	4 (36.4)	4 (36.4)	3 (27.3)	2 (33.3)	3 (50)	1 (16.6)	2 (66.6)	1 (33.3)	0	11 (10.3)	6 (7.6)	3 (7.7)
Breast + complementary	23 (63.8)	8 (22.2)	5 (13.8)	18 (64.3)	6 (21.4)	4 (14.3)	7 (43.8)	4 (25)	3 (18.8)	36 (33.6)	28 (35.4)	16 (41)
Solid food	16 (26.6)	26 (43.3)	18 (30)	12 (44.4)	20 (74.1)	13 (48.1)	3 (13.6)	11 (50)	8 (36.4)	60 (56.1)	27 (34.2)	22 (56.4)
Source of water												
Tap water	31 (42.5)	22 (30.1)	20 (27.4)	22 (42.3)	16 (30.8)	14 (26.9)	8 (33.3)	7 (29.2)	9 (37.5)	73 (68.2)	52 (65.8)	24 (61.5)
Well	10 (34.5)	14 (48.3)	5 (17.2)	9 (37.5)	11 (45.8)	4 (16.6)	2 (16.6)	8 (66.6)	2 (16.6)	29 (27.1)	24 (30.4)	12 (30.8)
Boiled	2 (40)	2 (40)	1 (20)	1 (33.3)	2 (66.6)	0	2 (66.6)	1 (33.3)	0	5 (4.7)	3 (3.8)	3 (7.7)
Contact with animals												
Yes	26 (45.6)	16 (28.1)	15 (26.3)	19 (46.3)	12 (29.3)	10 (21.4)	8 (36.4)	8 (36.4)	6 (27.3)	57 (53.3)	41 (51.9)	22 (56.4)
No	17 (34)	22 (44)	11 (22)	13 (34.2)	17 (44.7)	8 (21.1)	4 (23.5)	8 (47)	5 (29.4)	50 (46.8)	38 (48.1)	17 (43.6)

IGH: Injibara General Hospital, DPH: Dangila Primary Hospital, AGPH: Agew Gimjabet Primary Hospital, DEC: diarrheagenic *Escherichia coli*, Total^t^: total tested, Total^c^: total of culture-positive, Total^p^: total of PCR-positive.

**Table 3 tropicalmed-09-00065-t003:** Phenotypic multiple-antibiotic resistance in diarrheagenic *Escherichia coli*.

Pattern No.	Antibiotic-Resistant Patterns	No. of Antibiotics (Classes)	MDR Isolates
			n (%)
1 *	GEN, AMX	2 (2)	4 *
2 *	AMX, TMP/SMX	2 (2)	2 *
3	AMP, TMP/SMX, STR	3 (3)	4 (12.1)
4	TET, AMX, TMP/SMX	3 (3)	3 (9.1)
5	TMP, TET, GEN, TMP/SMX	4 (3)	6 (18.2)
6	GEN, AMX, TMP/SMX, STR	4 (3)	3 (9.1)
7	AMP, AMX, TMP/SMX, STR	4 (3)	4 (12.1)
8	TET, AMP, GEN, AMX, STR	5 (3)	3 (9.1)
9	TMP, AMP, AMX, GEN, STR	5 (3)	2 (6.1)
10	TMP, TET, AMP, GEN, AMX, STR	6 (4)	4 (12.1)
11	TMP, AMP, GEN, AMX, TMP/SMX, STR	6 (3)	3 (9.1)
12	TMP, TET, AMP, GEN, AMX, TMP/SMX, STR	7 (4)	1 (3)

MDR: multidrug-resistant, AMX: amoxicillin, AMP: ampicillin, GEN: gentamicin, STR: streptomycin, TMP: trimethoprim, TMP/SMX: trimethoprim/sulfamethoxazole, TET: tetracycline, * non-multidrug-resistant.

## Data Availability

All data generated or analyzed during this study are included in this manuscript.

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
