# Peer review of "Characteristics of Pathogenic Escherichia coli Associated with Diarrhea in Children under Five Years in Northwestern Ethiopia"

_tropicalmed, 2024, doi:10.3390/tropicalmed9030065_

Round 1

Reviewer 1 Report

Comments and Suggestions for Authors

The authors aimed to study the prevalence of diarrheagenic E. coli (DEC) in stool samples, from children under 5 years with diarrhea, from Ethiopia. Isolates were identified and characterized by a combination of culture and molecular methods, giving relevance to the virulence genes and antibiotic resistance pattern. Results showed a high prevalence of DEC, with ETEC as the predominant pathotype found. Moreover, DECs showed high resistance to antibiotics from several antibiotic classes, revealing multidrug-resistance.

The overall subject of the manuscript has scientific relevance, and fits the Tropical Medicine and Infectious Disease topics. In low-income countries, diarrheal diseases are the second-highest cause of childhood mortality and morbidity. Diarrhea accounts for over 10% of deaths in African children under five. E. coli is recognized a major etiological agent for childhood diarrhea. Moreover, E. coli integrates the WHO antibiotic-resistant “priority pathogen” list as priority 1 level (critical). There is still a lack of knowledge about the topic, particularly in low-income countries.

The authors showed a good command of the English language and overall the manuscript flow is good and the information is clear. The presentation needs to be polished. Minor editing of English language required.

Please see some suggestions for improvement below:

1.      Consider to change title to “Prevalence of pathogenic Escherichia coli associated with diarrhea in children under five years in Northwestern Ethiopia”

2.      English language edits throughout the manuscript must be addressed. Species names must be in italics. After the first time the species names do not need to be in full, revise throughout the manuscript.

3.      Line 13: Change “occurrence” to “prevalence”

4.      Line 14 and throughout the manuscript: Change “E. coli strains” to “E. coli pathotypes”.

5.      Line 16: Define DEC.

6.      Line 22: Change to “two to seven antibiotics classes”.

7.      Line 30: Remove “(E. coli)”.

8.      Line 44: EHEC is missing.

9.      Introduction: Give some information about the diarrheal relevance in African and Ethiopia. To showcase the relevance of the study is important to note the disease burden. In Africa were reported 114 479 diarrhea episodes per 100 000 person per year, and 576 592 deaths (2019). In Ethiopia the number of diarrhea episodes was 108 476, with 50 773 associated deaths (25 139 deaths in children with < 5 years). https://ourworldindata.org/grapher/incidence-of-diarrheal-diseases

10.   Add information in the introduction about the WHO list of antibiotic-resistant “priority pathogen” where E. coli is recognized as priority level 1 (critical).

11.   Line 35: Change “significant” to “relevant/important/key”.

12.   Line 55: Clearly define the study country.

13.   Figure 1. The figure is difficult to read and do not give essential information to the main reader, since no geographic features are displayed. The color brown in the bottom left figure is not in Legend 2. Consider to change the figure to include only a map with the geographic location of the country and the hospitals towns locations to the capital.

14.   Line 73: Add the definition of diarrhea episode used in this study.

15.   Line 85: Change “Eosin Methylene Blue” to “EMB”.

16.   Line 91: Define TSB.

17.   Line 93 – 111: Give the reagents concentrations, including primers. Revise the units and formats. Suggestion: You can add the annealing temperature for each primer/gene in the Table 1.

18.   Line 1115 – 119: Present the antibiotics evaluated by antibiotic class.

19.   Line 141/148: Table 1 is the primers table.

20.   Line 148: Remove “of positivity”.

21.   Table 2 footnote: Change “tasted” to “tested”.

22.   Line numbers restart on page 8.

23.   Line 5/page 8: Table 1 should be Table 2.

24.   Revise the formats for the genes enumeration throughout the manuscript.

25.   Perform statistical analysis to see if the differences found are significant.

26.   Line 16/page 8: Figure 2 is missing.

27.   Table 3: Revise the formats.

28.   Line 28/page 8: Remove the adjective “notably”.

29.   Line 31/page 8: Remove “(low prevalence of resistance)”.

30.   Figure 3: Group the antibiotics according the classes. Add the virulence genes is the heatmap.

31.   Page 10 and throughout the manuscript: Remove the apostrophe in strains and children. The use of the apostrophe is a literary resource and should not be used in scientific writing.

32.   Page 10/Line 54 – 56: Please rewrite, the isolates were identified as DEC based on the presence of the virulence genes. DEC was already defined in full.

33.   Since the E. coli percentages found are similar to the ones in this study, you can add the reference:

 Mero S, Timonen S, La¨a¨veri T, Løfberg S, Kirveskari J, Ursing J, et al. (2021) Prevalence of diarrhoeal pathogens among children under five years of age with and without diarrhoea in Guinea-Bissau. PLoS Negl Trop Dis 15(9): e0009709.  https://doi.org/10.1371/journal.pntd.0009709

34.   Line 78/page 11: Change “certain” to “specific”.

35.    Page 11: Give the name of the author and after insert the reference (e.g. Juste et al. [52]).

36.   Line 94/page 11: Change to “two virulence genes”.

37.   Give information about the main antibiotic prescribed in the country.

38. The Results and Discussion need to revise and rewrite at some degree to better showcase the information. Remove the repetitions and highlight the DEC/MDR occurrence.

39. Please address the study limitations.

Comments on the Quality of English Language

The authors showed a good command of the English language and overall the manuscript flow is good and the information is clear. The presentation needs to be polished. Minor editing of English language required.

Author Response

We would like to thank you very much for your constructive comments that improved the quality of the manuscript. We have carefully studied the comments and suggestions and revised our paper accordingly. The following are our point-by-point responses to the major and minor comments.

  1. Consider changing the title to “Prevalence of pathogenic Escherichia coliassociated with diarrhea in children under five years in Northwestern Ethiopia”

Response: We thank the reviewer for the comment. We have updated the paper's title to “Characteristics of pathogenic Escherichia coli associated with diarrhea in children under five years in Northwestern Ethiopia” based on the suggestion. The authors have chosen the term “characteristics” instead of “prevalence” since, overall, the aim of the paper was to characterize DEC pathotypes associated with intestinal infections, and the word prevalence is more commonly used in epidemiological studies.

  1. English language edits throughout the manuscript must be addressed. Species names must be in italics. After the first time the species names need not be in full, revise throughout the manuscript.

Response: The authors acknowledge the comment. The manuscript was checked, and corrections have been made throughout the manuscript accordingly.

  1. Line 13: Change “occurrence” to “prevalence”

Response: We thank the reviewer for the comments. This has been corrected in the revised manuscript (line 14).

  1. Line 14 and throughout the manuscript: Change “E. colistrains” to “E. coli pathotypes”.

Response: The authors acknowledge the comment. The correction has been made based on the comments throughout the manuscript.

  1. 5. Line 16: Define DEC.

Response: Done according to the suggestions. The correction has been made on line 12.

  1. Line 22: Change to “two to seven antibiotics classes”.

Response: Modification has been done accordingly (lines 26–27).

  1. Line 30: Remove “(E. coli)”.

Response: This has been carried out as per the comments (line 34)

  1. Line 44: EHEC is missing.

Response: Thank you. We have included the aforementioned pathotype properly (lines 45–46).

  1. Introduction: Give some information about the diarrheal relevance in Africa and Ethiopia. To showcase the relevance of the study is important to note the disease burden. Africa reported 114 479 diarrhea episodes per 100,000 person per year, and 576 592 deaths (2019). In Ethiopia, the number of diarrhea episodes was 108 476, with 50 773 associated deaths (25 139 deaths in children with < 5 years). https://ourworldindata.org/grapher/incidence-of-diarrheal-diseases.

Response: We acknowledge the comment. A major revision of the introduction section has been carried out as per the comments. Please find these in the revised manuscript, lines 63–76.

  1. Add information in the introduction about the WHO list of antibiotic-resistant “priority pathogens” where E. coliis recognized as priority level 1 (critical).

Response: Thank you. We have added information in the introduction section according to the comments and suggestions (79–81).

  1. Line 35: Change “significant” to “relevant/important/key”.

Response: Modification has been done accordingly.

  1. Line 55: Clearly define the study country.

Response: Done according to the suggestions (line 87).

  1. Figure 1. The figure is difficult to read and does not give essential information to the main reader, since no geographic features are displayed. The color brown in the bottom left figure is not in Legend 2. Consider changing the figure to include only a map with the geographic location of the country and the hospital's town's location to the capital.

Response: Thank you. The map of the study area is updated based on the comments (Figure 1).

  1. Line 73: Add the definition of diarrhea episode used in this study.

Response: The definition of acute diarrhea has been added appropriately (113–115).

  1.  Line 85: Change “Eosin Methylene Blue” to “EMB”.

Response: Done accordingly. The correction has been made on line 123.

  1. Line 91: Define TSB.

Response: Done accordingly (line 130).

Line 93 – 111: Give the reagents concentrations, including primers. Revise the units and formats. Suggestion: You can add the annealing temperature for each primer/gene in Table 1.

Response: We thank you and agree with the comments. The concentrations of each reagent are provided, and the forwarded suggestion is accepted (lines 133–143; Table 1).

  1. Line 115 – 119: Present the antibiotics evaluated by the antibiotic class.

Response: Thank you. The comments indicated are considered and have been modified accordingly (lines 153–157).

  1.  Line 141/148: Table 1 is the primers table.

Response: I apologize for this inconsistency and thank you for the notification. This has been corrected in the revised manuscript (line 178).

Line 148: Remove “of positivity”.

Response: The deletion has been done accordingly (line 182).

  1. Table 2 footnote: Change “tasted” to “tested”.

Response: The modification is done based on the comment (Table 2).

  1. Line numbers restart on page 8.

Response: Thank you. The issue is caused by a page break.

  1. Line 5/page 8: Table 1 should be Table 2.

Response: Apologize for this inconsistency and thank you for the notification. This has been corrected in the revised manuscript (line 7, after the page break).

  1. Revise the formats for the gene enumeration throughout the manuscript.

Response: We have revised the formats of gene enumeration throughout the manuscript accordingly.

  1. Perform statistical analysis to see if the differences found are significant.

Response: Thank you for your comment. However, the sampling procedure used in this study is inappropriate for statistical analyses.

  1. Line 16/page 8: Figure 2 is missing.

Response: Figure 2 is added to the result section of the manuscript by reformatting Table 3 based on the suggestions.

  1. Table 3: Revise the formats.

Response: The presentation format has been revised and changed to figures for better understanding and attractiveness (Figure 2).

  1. Line 28/page 8: Remove the adjective “notably”.

Response: Done according to the suggestions (line 33).

  1. Line 31/page 8: Remove “(low prevalence of resistance)”.

Response: Done accordingly (line 36).

  1. Figure 3: Group the antibiotics according to the classes. Add the virulence genes to the heatmap.

Response: The modification has been carried out as per the comments. Please find these in the revised manuscript in Figure 3.

  1. Page 10 and throughout the manuscript: Remove the apostrophe in strains and children. The use of the apostrophe is a literary resource and should not be used in scientific writing.

Response: Thank you. We have revised it according to the comments and suggestions throughout the manuscript.

  1. Page 10/Line 54 – 56: Please rewrite, the isolates were identified as DEC based on the presence of the virulence genes. DEC was already defined in full.

Response: The correction has been done based on the comments (line 65).

  1. Since the E. colipercentages found are similar to the ones in this study, you can add the reference:

 Mero S, Timonen S, La¨a¨veri T, Løfberg S, Kirveskari J, Ursing J, et al. (2021) Prevalence of diarrhoeal pathogens among children under five years of age with and without diarrhoea in Guinea-Bissau. PLoS Negl Trop Dis 15(9): e0009709.  https://doi.org/10.1371/journal.pntd.0009709

Response: Thank you. The reference is added accordingly (line 69).

  1. Line 78/page 11: Change “certain” to “specific”.

Response: Done accordingly (line 120).

  1. Page 11: Give the name of the author and insert the reference (e.g. Juste et al.[52]).

Response: The modification has been done in the discussion section of the manuscript (lines 132–164).

  1. Line 94/page 11: Change to “two virulence genes”.

Response: Done according to the suggestions (line 141).

  1. Give information about the main antibiotic prescribed in the country.

Response: The information has been added based on the comment (lines 160–162).

  1. The Results and Discussion need to be revised and rewritten to some degree to better showcase the information. Remove the repetitions and highlight the DEC/MDR occurrence.

Response: We have revised the result and discussion profoundly according to the comments and suggestions.

  1. Please address the study limitations.

Response: Thank you for your constructive comments and suggestions. The limitations of the study have been addressed accordingly in the revised manuscript (lines 189–198).

Reviewer 2 Report

Comments and Suggestions for Authors

The Authors describe an interesting survey on pathogenic Escherichia coli among 107 children under five years with diarrhea observed in three hospitals of the Awi Ethiopian administrative region. I suggest a major revision beacuse lot of data (fruit of the worthy and in-depth study carried out) could be really more valued and discussed in the final chapter.

Following my comments and questions:

Line 15: in addition to the number of total children studied, specify the number of children with Escherichia coli and of children with pathogenic E. coli (DEC).

line 14: I suggest to change the word "strains" in “pathotypes” for avoiding confusion between the isolates effectively observed (79) and the combination of virulence genes (6) detected among them;

Lines 16 and 18: display the acronyms DEC, ETEC, etc,  as they are used for the first time in the text (copy lines 40-41-44);

Line 22: speaking about the MDR strains, it’s better referring not to the antibiotics but to the categories of antibiotics for which E.coli are found resistant

Line 31: replace “microbiota” (a community of microrganisms) with “microrganism”;

Line 31 His ability…makes it…. (you are speaking generically about Escherichia coli as single species and not about different strains of coli);

Line 41: I suggest (if accepted, remember to put  in lines 101-102 and page 8 only the acronyms of genes) to insert here a brief description of different virulence genes with its role in the pathogenic action of E. coli.

Figure 1: Map 1: make evident the neighboring countries (Somalia, Kenia, etc) for better understanding where Ethiopia is placed in Africa continent; Map 2: remove the highlight of the yellow region as it’s not the area of study; Map 3: highlight with different colors the three towns (the names of two towns in the map are covered from black points and not readables).

Line 79: change “demographics” with “anamnesis” intending here, I suppose, not only the geographical origin but also age, feeding, etc as reported in Table 2;

Line 95-96-97: correct µ1 in µl;

Lines 101-102: display here the acronyms of virulence gene and use only the acronyms in page 8 (lines 6-9).

Line 116: …different molecules (or antibiotics);

Line 120: …”following” Clinical…guidelines

Line 145: specify here exactly the number, with percentages, of pathogenic E.coli isolated (and not all E.coli isolated) in the three hospitals as reported in Table 2 because the DEC are considered from the Author the cause of diarrhea in these 39 children (and not a generic E.coli infection; if it’s not, it becomes mandatory discuss the role of not pathogenic Escherichia coli for the other cases of diarrhea).

Table 1: write extensively CF (complementary food) like the other terms in the table.

From Page 8:

lines 6-9: see comment line 101-102; insert here only the acronym.

Line 12: not EPEC but aEPEC; beacuse the Author insert here abruptly this type of pathogenic E. coli, I suggest to insert in introduction (line 42) a reference to aEPEC and tEPEC and why these are different (see also comment line 41);

Table 3: theoretically EHEC with stx2 would be STEC coli (here they appear separated); as some authors have a different approach to classification of EHEC and STEC on the basis of virulence gene combination, insert a bibliographic reference that support your classification of the pathogenic coli isolated in the study;

Table 3: it lacks the total number of EHEC (3).

Line 34: as in the abstract (line 22), emphasize in the text not the number of antibiotics but the number of antimicrobial categories for which the E. coli were found resistant.

Figure 3: the data of this Figure are not discussed, consider deleting it; see also the comment about Discussion.

Discussion: the Authors have recovered a big quantity of interesting data that would be worthy of being considered (for example, the distribution of antimicrobial susceptibility profiles in different hospital in Figure 3 probably a reflection of local different therapeutic habits by doctors?; the reasons for which solid food is an evident risk factor for acquiring pathogenic E. coli as compared to mother’s milk – maternal immunity?; the big topic of the water and the contact with animals, with many studies in  the world about the diffusion of pathogenic E. coli among them; etc).

Line 50 and 51: as in line 14, E.coli pathotypes and not strains.

Line 57: which countries?

Line 65: if it does exist bibliographic references, compare your results not only with India but also with some other countries.

Line 60 and 72: see also comment lines 41 and 12 (after page 8) about the description of aEPEC and tEPEC

Line 60: about the STEC, have been observed in the carrier children symptoms of classical HUS such as anuria, oliguria, kidney damage, anemia, etc?

Line 73: here and throughout all the Discussion chapter (lines 84,  90, 92, 113,  etc) insert the name of the first Author of paper considered and not only the number of reference (for example:"...in line with those of Borujerdi et al. (30)").

Line 84: reported by who and where?

Line 97: In which countries these studies have been made?

Line 112: see comments above (line 22 and over)

Line 126: as "Conclusions" is a generic chapter, anticipate the new acronym MRDEC in line 108 or over in Discussion chapter.

Line 221: eliminate Wayne P (it s not the author of CLSI guidelines).

Author Response

We would like to thank you very much for your constructive comments that improved the quality of the manuscript. We have carefully studied the comments and suggestions and revised our paper accordingly. The following are our point-by-point responses to the major and minor comments.

Line 15: in addition to the number of total children studied, specify the number of children with Escherichia coli and of children with pathogenic E. coli (DEC).

Response: Thank you for the comment. The abstract has been corrected accordingly in the revised manuscript (lines 15–20).

line 14: I suggest changing the word "strains" to “pathotypes” to avoid confusion between the isolates effectively observed (79) and the combination of virulence genes (6) detected among them;

Response: The correction has been done based on the comments throughout the manuscript.

Lines 16 and 18: display the acronyms DEC, ETEC, etc, as they are used for the first time in the text (copy lines 40-41-44);

Response: We have revised it according to the comments and suggestions. The correction has been made on lines 20–22.

Line 22: speaking about the MDR strains, it’s better to refer not to the antibiotics but to the categories of antibiotics for which E. coli are found resistant

Response: The correction has been done based on the comments (lines 25–27).

Line 31: replace “microbiota” (a community of microorganisms) with “microorganism”;

Response: Done accordingly (line 35).

Line 31 His ability…makes it…. (you are speaking generically about Escherichia coli as a single species and not about different strains of coli);

Response: Modification has been done accordingly (lines 35–36).

Line 41: I suggest (if accepted, remember to put in lines 101-102 and page 8 only the acronyms of genes) to insert here a brief description of different virulence genes with their role in the pathogenic action of E. coli.

Response: We have revised and added the requested information to the revised manuscript, particularly in the introduction part of the paper (lines 47–62).

Figure 1: Map 1: make evident the neighboring countries (Somalia, Kenia, etc) for better understanding where Ethiopia is placed in Africa continent; Map 2: remove the highlight of the yellow region as it’s not the area of study; Map 3: highlight with different colors the three towns (the names of two towns in the map are covered from black points and not readables).

Response: Response: Thank you. The map of the study area is updated based on the comments (Figure 1).

 Line 79: change “demographics” with “anamnesis” intending here, I suppose, not only the geographical origin but also age, feeding, etc as reported in Table 2;

Response: Done according to the suggestions. The correction has been made on line 117.

Line 95-96-97: correct µ1 in µl;

Response: It is done accordingly

Lines 101-102: display here the acronyms of the virulence gene and use only the acronyms on page 8 (lines 6-9).

Response: Modification has been done accordingly.

Line 116: …different molecules (or antibiotics);

Response: The correction has been made to lines 152-153.

Line 120: …”following” Clinical…guidelines

Response: Modification has been done accordingly (line 158).

Line 145: specify here exactly the number, with percentages, of pathogenic E. coli isolated (and not all E.coli isolated) in the three hospitals as reported in Table 2 because the DEC are considered from the Author the cause of diarrhea in these 39 children (and not a generic E.coli infection; if it’s not, it becomes mandatory discuss the role of not pathogenic Escherichia coli for the other cases of diarrhea).

Response: The authors acknowledge the comment. The correction has been done based on the comments (lines 3-5, after the page break.)

Table 1: Write extensively CF (complementary food) like the other terms in the table.

Response: Done accordingly (Table 2)

lines 6-9: see comment lines 101-102; insert here only the acronym.

Response: The correction has been done based on the comments.

Line 12: not EPEC but aEPEC; because the Author insert here abruptly this type of pathogenic E. coli, I suggest inserting in the introduction (line 42) a reference to aEPEC and tEPEC and why these are different (see also comment line 41);

Response: We thank the reviewer for the comments. The comments indicated are considered and have been modified accordingly (line 16, after page break) and (53-58, before page break)

Table 3: theoretically EHEC with stx2 would be STEC coli (here they appear separated); as some authors have a different approach to the classification of EHEC and STEC on the basis of virulence gene combination, insert a bibliographic reference that support your classification of the pathogenic coli isolated in the study;

Response: Thank you. The criteria for the classification of EHEC and STEC pathotypes are added in the introduction section of the paper (lines 58–60).

Table 3: it lacks the total number of EHEC (3).

Response: The correction has been made in revised figure 2

Line 34: as in the abstract (line 22), emphasize in the text not the number of antibiotics but the number of antimicrobial categories for which the E. coli were found resistant.

Response: Thank you. The issue has been corrected accordingly in the revised manuscript.

Figure 3: the data of this Figure are not discussed, consider deleting it; see also the comment about Discussion.

Response: The data in Figure 3 are discussed properly in the revised manuscript and it gives clear information on the distribution of antimicrobial and virulence profiles in the study area. Hence, we believe that the presence of Figure 3 would enhance the quality of the manuscript.

Discussion: the Authors have recovered a big quantity of interesting data that would be worthy of being considered (for example, the distribution of antimicrobial susceptibility profiles in different hospital in Figure 3 probably a reflection of local different therapeutic habits by doctors?; the reasons for which solid food is an evident risk factor for acquiring pathogenic E. coli as compared to mother’s milk – maternal immunity?; the big topic of the water and the contact with animals, with many studies in  the world about the diffusion of pathogenic E. coli among them; etc).

Response: The authors are thankful for your acknowledgment. We also appreciate the reviewer's perspective on the study. The indicated suggestions are considered, and we have added the aforementioned perspectives in the discussion part of the manuscript (lines 71–87; 88–100; 146–156).

Line 50 and 51: as in line 14, E. coli pathotypes and not strains.

Response: Modifications are done accordingly (lines 58–63)

Line 57: which countries?

Response: The countries are listed in lines 69–70.

Line 65: if it does exist bibliographic references, compare your results not only with India but also with some other countries.

Response: The modification has been done accordingly.

Line 60 and 72: see also comment lines 41 and 12 (after page 8) about the description of aEPEC and tEPEC

Response: The descriptions of aEPEC and tEPEC are included in the introduction section of the revised manuscript (lines 55–58).

Line 60: about the STEC, have been observed in the carrier children symptoms of classical HUS such as anuria, oliguria, kidney damage, anemia, etc?

Response: We have not considered those symptoms of HUS in the children because the primary objective of the study is focused on the intestinal pathogenic E. coli that causes diarrhea.

Line 73: here and throughout all the Discussion chapter (lines 84, 90, 92, 113,  etc) insert the name of the first Author of paper considered and not only the number of reference (for example:"...in line with those of Borujerdi et al. (30)").

Response: We appreciate the comments. The comments indicated are considered and have been modified accordingly (lines 133–175).

Line 84: reported by who and where?

Response: The authors and the countries of the studies that were used for discussion are added based on the comments (lines 125–126).

Line 97: In which countries these studies have been made?

Response: Modification has been done accordingly (lines 144–145).

Line 112: see comments above (line 22 and over)

Response: Corrections are carried out based on the comments throughout the manuscript.

Line 126: as "Conclusions" is a generic chapter, anticipate the new acronym MRDEC in line 108 or over in the Discussion chapter.

Response: Thank you. A major revision of the conclusion and discussion part of the revised manuscript is done based on the comments.

Line 221: eliminate Wayne P (it’s not the author of CLSI guidelines).

Response: Done as per the comment.

Reviewer 3 Report

Comments and Suggestions for Authors

Mossie and colleagues conducted an investigation into the frequency of six diarrheal pathotypes of Escherichia coli in children <5 years of age in the Awi Zone, Northwest Ethiopia. The paper is generally well written but some of the information is either incomplete or confusing. As an aside, I note that the first author's name here is listed as Beirhun Mossie Mulu, while on the printed copy I reviewed it is Beihun Mossie only. 

A few additions and clarifications would improve the clarity of this paper:

1) In the Abstract, the authors report that they isolated E. coli from 107 diarrheal stool samples, while on line 139 they state that 107 stool samples were collected and then on line 143 (page 5 of 15) they report that E. coli was recovered from 79 samples. Which is it?

2) Given that diarrhea is very common in the region under study, why only 107 stool samples over 7 months? Were these from hospitalized children or any child presenting with diarrhea to the three hospitals?

3) Of the 79 Escherichia coli isolates, only 39 had detectable virulence genes (Section 3.2, line 2). Thus 40 isolates are of questionable relevance in terms of pathogenicity of the diarrhea.

4) The authors never mention what else was recovered from the stool cultures. Co-pathogens are common in stools, more so in developing regions of the world.

5) The data of antimicrobial resistance in the PCR positive Escherichia coli isolates is very important. However, some clinical context would be helpful. How many of the 39 such children got antibiotics? How many received antibiotics to which E. coli was resistant and recovered anyway? How many children with clinical failures had co-pathogens?

Author Response

We would like to thank you very much for your constructive comments that improved the quality of the manuscript. We have carefully studied the comments and suggestions and revised our paper accordingly. The following are our point-by-point responses to the major and minor comments.

1) In the Abstract, the authors report that they isolated E. coli from 107 diarrheal stool samples, while on line 139 they state that 107 stool samples were collected, and then on line 143 (page 5 of 15), they report that E. coli was recovered from 79 samples. Which is it?

Response: Thank you for the comment. The abstract has been corrected accordingly in the revised manuscript (lines 15–19).

2) Given that diarrhea is very common in the region under study, why were only 107 stool samples over 7 months? Were these from hospitalized children or any child presenting with diarrhea to the three hospitals?

Response: The authors acknowledge the comment. We have clearly stated in the methodology of the study that the study subjects are children with diarrhea who visited those three hospitals due to acute diarrhea but were not hospitalized. Although diarrhea is common in the area, seeking medical attention and visiting hospitals for the management of the condition is unusual. In addition, the exclusion criteria, which include antibiotic therapy, an age older than five years, and diarrhea lasting longer than 14 days, resulted in a reduced number of samples for the study.

3) Of the 79 Escherichia coli isolates, only 39 had detectable virulence genes (Section 3.2, line 2). Thus 40 isolates are of questionable relevance in terms of the pathogenicity of the diarrhea.

Response: Yes, indeed. The 39 isolates that harbor virulence genes are responsible for diarrhea. 

4) The authors never mention what else was recovered from the stool cultures. Co-pathogens are common in stools, more so in developing regions of the world.

Response: We thank the reviewer for the comment. However, it is important to emphasize that the primary objective of this study was to investigate the prevalence of six DEC pathotypes in children with diarrhea and determine their antibiotic resistance patterns. Due to resource constraints, monitoring multiple pathogens at this level of study is challenging. However, we acknowledge this as one of our study limitations (lines 191–193).

5) The data of antimicrobial resistance in the PCR-positive Escherichia coli isolates is very important. However, some clinical context would be helpful. How many of the 39 such children got antibiotics? How many received antibiotics to which E. coli was resistant and recovered anyway? How many children with clinical failures had co-pathogens?

Response: Thank you. However, all children who received antibiotic therapy within the previous two weeks were excluded from this study. Hence, it does not apply to the prior suggestion in the current study.

Round 2

Reviewer 1 Report

Comments and Suggestions for Authors

In the revised version of the manuscript, the authors addressed or responded to most comments previously made.

Comments on the Quality of English Language

Minor editing of English language required that can be done during the proofread phase.

Reviewer 2 Report

Comments and Suggestions for Authors

Thanks to the Authors for revising the manuscript as suggested; consider these last two corrections: 1) line 5 after the page break, round up the prevalence 30.8% in 31% as the total prevalence in the three hospital must be equal to 100%; 2) line 317 eliminate one of the double (32).